# Circulating Immune Proteins: Improving the Diagnosis and Clinical Outcome in Advanced Non-Small Cell Lung Cancer

**DOI:** 10.3390/ijms242417587

**Published:** 2023-12-18

**Authors:** Susana Torres-Martínez, Silvia Calabuig-Fariñas, Sandra Gallach, Marais Mosqueda, Ester Munera-Maravilla, Rafael Sirera, Lara Navarro, Ana Blasco, Carlos Camps, Eloisa Jantus-Lewintre

**Affiliations:** 1Molecular Oncology Laboratory, Fundación Investigación Hospital General Universitario de Valencia, 46014 Valencia, Spain; sutormar@doctor.upv.es (S.T.-M.); gallach_sangar@gva.es (S.G.); mosqueda_mar@gva.es (M.M.); munera_est@gva.es (E.M.-M.); camps_car@gva.es (C.C.); ejantus@btc.upv.es (E.J.-L.); 2TRIAL Mixed Unit, Centro Investigación Príncipe Felipe—Fundación Investigación Hospital General Universitario de Valencia, 46014 Valencia, Spain; blasco_ana@gva.es; 3Centro de Investigación Biomédica en Red Cáncer, CIBERONC, 28029 Madrid, Spain; 4Department of Pathology, Universitat de València, 46010 Valencia, Spain; 5Department of Biotechnology, Universitat Politècnica de València, 46022 Valencia, Spain; rsirera@btc.upv.es; 6Department of Pathology, Hospital General Universitario de Valencia, 46014 Valencia, Spain; navarro_larcera@gva.es; 7Department of Medical Oncology, Hospital General Universitario de Valencia, 46014 Valencia, Spain; 8Department of Medicine, Universitat de València, 46010 Valencia, Spain; 9Nanomedicine, Centro Investigación Príncipe Felipe—Universitat Politècnica de Valencia, 46022 Valencia, Spain

**Keywords:** NSCLC, pembrolizumab, plasma, circulating immune proteins, predictive, prognostic

## Abstract

Immunotherapy has been proven a viable treatment option for non-small cell lung cancer (NSCLC) treatment in patients. However, some patients still do not benefit. Finding new predictive biomarkers for immunocheckpoint inhibitor (ICI) response will improve treatment management in the clinical routine. In this regard, liquid biopsy is a useful and noninvasive alternative to surgical biopsies. In the present study, we evaluated the potential diagnostic, prognostic, and predictive value of seven different soluble mediators involved in immunoregulation. Fifty-two plasma samples from advanced NSCLC treated in first-line with pembrolizumab at baseline (PRE) and at first response assessment (FR) were analyzed. In terms of diagnostic value, our results revealed that sFGL1, sGAL-3, and sGAL-1 allowed for optimal diagnostic efficacy for cancer patients. Additionally, the combination of sFGL1 and sGAL-3 significantly improved diagnostic accuracy. Regarding the predictive value to assess patients’ immune response, sCD276 levels at PRE were significantly higher in patients without tumor response (*p* = 0.035). Moreover, we observed that high levels of sMICB at PRE were associated with absence of clinical benefit (pembrolizumab treatment less than 6 months) (*p* = 0.049), and high levels of sMICB and sGAL-3 at FR are also related to a lack of clinical benefit (*p* = 0.027 and *p* = 0.03, respectively). Finally, in relation to prognosis significance, at PRE and FR, sMICB levels above the 75th percentile are related to poor progression-free survival (PFS) (*p* = 0.013 and *p* = 0.023, respectively) and overall survival (OS) (*p* = 0.001 and *p* = 0.011, respectively). An increase in sGAL3 levels at FR was associated with worse PFS (*p* = 0.037). Interestingly, high sGAL-3 at PRE was independently associated with PFS and OS with a hazard ratio (HR) of 2.45 (95% CI 1.14–5.25; *p* = 0.021) and 4.915 (95% CI 1.89–12.73; *p* = 0.001). In conclusion, plasma levels of sFGL1, sGAL-3, and sGAL-1 could serve as diagnostic indicators and sMICB, sCD276, and sGAL3 were linked to outcomes, suggesting their potential in assessing NSCLC under pembrolizumab treatment. Our results highlight the value of employing soluble immune biomarkers in advanced lung cancer patients treated with pembrolizumab at first-line.

## 1. Introduction

Lung cancer is the leading cause of cancer death worldwide [1]. The 5-year survival rate for lung cancer patients is 18% due to most patients being diagnosed at advanced stages. Currently, various techniques are accessible to establish a conclusive diagnosis. Significant improvements in diagnostic procedures are being made to optimize imaging acquisition and tumor localization; however, it remains inadequate in some cases due to difficulties accessing the tumor [2]. Tumor development and progression is a multistage process that involves immune system. A compromised immune system is a distinctive feature of solid tumors. Exploring the role of the immune system in the development and progression of lung cancer has emerged as a focal point in research over the past few decades. Based on this type of research, there is currently an effort to develop cancer immunotherapies that specifically address the immune microenvironment within tumors [3]. Advances in immunotherapy have led to a new standard of treatment for metastatic non-small cell lung cancer (NSCLC). Specifically, many studies have been performed around the programmed death 1 (PD1) and programmed death ligand 1(PD-L1) axis inhibitors with plenty of results confirming better outcomes compared to conventional therapies [4,5,6,7]. Despite tumors’ ability to escape from immunosurveillance by expression of PD-L1 within the tumor microenvironment, there is renewed hope in the field of cancer immunotherapy with the promising approach of using immune checkpoint inhibitors (ICIs) to reinvigorate this antitumor immunity [8,9,10]. Despite initial effectiveness, many patients develop resistance to anti-PD-1/PD-L1 and anti-CTLA-4 inhibitors. This has led to the exploration of new immune checkpoint targets, which hold promise as therapeutic targets and are currently being clinically investigated [11].

ICIs have shown encouraging results during clinical development; regrettably, several patients remain refractory to these therapies and, therefore, the identification of patients who will derive benefit is a priority for treatment optimization. Approved predictive biomarkers, such as tumor PD-L1 expression, microsatellite instability (MSI), and the last tumor mutation burden (TMB), have been associated with higher response rates accompanied by extended overall survival (OS) across several cancer types [12,13,14,15]. To date, tumor PD-L1 expression assessed using immunohistochemistry (IHC) appears to be the best determinant of responsiveness to checkpoint blockades in NSCLC [5,6]. Indeed, the US Food and Drug Administration (FDA) has approved several immune checkpoint therapies linked to different PD-L1 HIC expression testing. As an example, pembrolizumab, an anti-PD-L1 antibody, is currently approved for first-line in patients whose tumors express ≥50% of PD-L1 and for second-line use in NSCLC patients whose tumors express at least 1% of PD-L1 without ALK or EGFR alterations [16]. However, PD-L1 expression, used in routine practice, as a biomarker, shows some limitations. Several studies have demonstrated a negative correlation between PD-L1 expression and response to checkpoint inhibitors. Some patients with high levels of tumor PD-L1 manifest poor responses and vice versa; patients with low or even negative PD-L1 expression exhibit good and durable responses to checkpoint inhibitors [4,17,18,19]. These divergent results may be explained by the heterogeneous, plastic, and dynamic expression of this marker as well as by variability in detection methods and lack of standardized criteria and cutoffs to determine positivity [19,20,21].

Considering these limitations, the requirement of new reliable biomarkers to lead patient selection and provide indications of efficacy has become critical. An ideal biomarker would preferably be minimally invasive. In this context, liquid biopsy is an alternative or complement to tissue biopsy. Nowadays, there is intense interest in identifying predictive biomarkers derived from peripheral blood samples, such as plasma or serum among others. In contrast to tumor specimens, plasma samples have the advantages of being accessible and allowing for clinical monitoring of therapy response. Some plasmatic biomarkers such as circulating tumor DNA (ctDNA) are currently used as an alternative to tumor tissues for mutation detection and companion diagnostic purposes [22,23,24]. In fact, the use of ctDNA enables guided treatment decisions for predicting response and resistance to targeted therapies and immunotherapies [25,26,27]. Nevertheless, a more comprehensive approach is needed to enable the analysis of interactions within the tumor microenvironment. Circulating immune proteins secreted by tumor cells or immune cells are involved in various biological functions, influence innate and adaptative immune function, and could be an important source of cancer biomarkers [28]. Specific proteins have been used for differential diagnosis, simplifying the identification of a particular type of lung cancer [3]. The use of circulating immune proteins in NSCLC to predict clinical outcomes has been sparsely validated in prospective studies and its role is not clearly understood. We previously studied the prognostic and predictive role of sGalectin-3 (sGAL-3), an immunosuppressive factor in the tumor microenvironment (TME), in a cohort of lung adenocarcinoma patients [29]. Based on the literature, proteins such as inducible t-cell costimulatory ligand (sICOSL), cluster of differentiation 276 (sCD276), fibrinogen-like protein 1 (sFGL1), galectin-1 (sGAL-1), human MHC class I polypeptide-related sequence A (sMICA), and human MHC class I polypeptide-related sequence B (sMICB) could play an important role as immunosuppressive or immune costimulatory proteins in the tumor microenvironment, but their role in lung cancer outcomes has not yet been elucidated [30,31,32,33]. For this reason, in the current study, we aimed to examine these seven new immune-related soluble markers (sFGL1, sICOSL sCD276, sGAL-1, sGAL-3, sMICA, and sMICB) in a cohort of NSCLC patients (including adenocarcinoma, squamous cell carcinoma, and other less common histologies), derived from plasma samples of NSCLC patients treated with pembrolizumab in first-line at baseline (PRE) and first response assessment (FR), to find novel potential diagnostic, predictive, and prognostic biomarkers for immunotherapy in NSCLC. This is the first complete study in which we analyze the diagnostic, prognostic, and predictive significance of seven different soluble mediators involved in immunoregulation in plasma samples from advanced NSCLC with first-line pembrolizumab treatment at PRE and FR.

## 2. Results

### 2.1. Clinical Characteristics of NSCLC Patients and Biomarker Assessment

Fifty-two advanced-stage NSCLC patients who received pembrolizumab as a first-line treatment were enrolled in this study. Table 1 summarizes the clinical characteristics. Patients were mostly male (75%), current smokers (71.2%), and with adenocarcinoma histology (65.4%). Patients showed good performance status (PS) (0-1) at pembrolizumab initiation in 84.6% of cases. None of the patients harbored targetable drivers approved by the European Medicines Agency (EMA). In accordance with the guidelines, PD-L1 expression ≥50% was present in tumor samples from all patients treated with pembrolizumab in monotherapy [34].

The objective response rate (ORR) was 40.4% (21/52). Twenty-nine patients presented durable clinical benefit (DCB) (four complete response, fifteen partial response, and ten stable disease), whereas the remaining twenty-three patients did not achieve DCB (non-DCB). Two of the patients had a partial response at the first-response assessment but progressed before 6 months. The median duration of follow-up was 18.41 months (range: 1.37–21.1912 months). Thirty-eight patients (73.1%) had progressed at data cutoff with a median progression-free survival (PFS) of 7 months (range: 0.1–13.36 months). Detailed patient characteristics and outcomes are shown in Appendix A.

Median levels of different analytes at pembrolizumab PRE and FR (4 months of treatment) can be found in Table 2. No significant differences were found between PRE and FR biomarker levels.

### 2.2. Correlations between Clinical Characteristics of NSCLC Patients and Biomarker Assessment

The significant correlations between the plasma levels of biomarkers at baseline and the clinical features of the patients are shown in Figure 1. As can be seen in the figure, at baseline, patients over 70 years old had high levels of sICOSL. Interestingly, former or current smokers exhibited high levels of coinhibitory checkpoints, sICOSL, sCD276, and sMICA, compared with nonsmokers. No correlations were found between plasma biomarker levels and sex, histology, stage, and PDL1 levels.

### 2.3. Biomarkers with Diagnostic Value

The median plasma biomarker levels of NSCLC patients were significantly higher than the controls (Figure 2).

ROC analysis was performed to test the ability of the potential biomarkers to diagnose NSCLC. The measurements of the different individual markers and their predictive values in the diagnosis of NSCLCs are summarized in Table 3. sFGL1, sGAL-3, and sGAL-1 were the biomarkers with the best overall diagnostic accuracy in the test. Among three plasma biomarkers, FGFL1 displayed the highest area under the curve (AUC) (0.919; 95% CI: 0.860–0.978), followed by sGAL-3 (AUC = 0.889; 95% CI: 0.827–0.960) and sGAL-1 (AUC = 0.801; 95% CI: 0.709–0.894) (Figure 3). A logistic regression was used to explore whether combining two or three plasma biomarkers would improve diagnostic accuracy. The combination of sFGL1 and sGAL-3 yielded a better optimal diagnostic efficacy for cancer patients (AUC = 0.963; 95% CI: 0.929–0.996) than the individual biomarker with a sensitivity of 82.7%, a specificity of 97.1%, a positive predictive value (PPV) of 99.7%, and a negative predictive value (NPV) of 78.5% to predict advanced cancer (Table 3).

### 2.4. Correlation of Plasma Biomarkers Levels with Response (ORR and Clinical Benefit) and Survival

#### 2.4.1. sCD276

Median sCD276 baseline levels were significantly higher in patients without tumor response with a median value of 874.05 pg/mL (IQR: 399.52–1306.96) compared to 326.38 pg/mL (IQR: 206.40–696.52) in patients with tumor response (*p* = 0.035) (Figure 4). The ORR was 26.9% (n = 7) in the case of those with high levels of sCD276 (n = 26) (≥median of 611.7850 pg/mL) versus 53.8% (n = 14) in the case of those with low levels of sCD276 (n = 26) (<median of 611.7850 pg/mL) (*p* = 0.048). However, there were no statistical differences at the first response assessment between patients who responded and those who did not. No statistical differences were found in sCD276 baseline levels in patients with DCB compared to non-DCB. There were also no differences found in first response assessment (Appendix A).

To perform the prognosis analysis, patients were stratified by the median value of sCD276 at PRE (611.41 pg/mL). Survival analysis showed that patients with low sCD276 levels tended to have better PFS and OS than those with high sCD276 levels (13.53 vs. 4.77 months, *p* = 0.055; 31.27 vs. 11.53 months, *p* = 0.120, respectively) (Appendix A). No significant results were found at the FR with the median value as cutoff (Appendix A). We also did not find significant results with the 75th percentile as cutoff.

#### 2.4.2. sMICB

When sMICB was analyzed and its potential value as predictive biomarker determined, at baseline, median sMICB levels were significantly higher in patients with non-DCB with a median value of 832.81 pg/mL (IQR: 590.38–1124.53) compared to 484.52 pg/mL (334.50–743.72) in patients with DCB (*p* = 0.049) (Figure 5A). Similarly, at first response assessment under pembrolizumab, sMICB levels were significantly higher in patients with non-DCB with a median value of 777.35 pg/mL (IQR: 486.55–938.08) compared to 446.73 pg/mL (348.31–555.70) in patients with DCB (*p* = 0.027) (Figure 5B). The clinical benefit rate (CBR) was 40% (n = 8) in cases of those with high levels of sMICB (n = 20) versus 80% (n = 16) in cases of those with low levels of sMICB (n = 20) (*p* = 0.022). However, there were no statistical differences in sMICB levels in patients who were responders compared to non-responders to pembrolizumab (Appendix A).

Using ROCs, sMICB cutoff levels of 583.39 pg/mL were associated with a sensitivity of 75%, a specificity of 79.2%, a positive predictive value of 70.6% and a negative predictive value of 82.6% to predict durable clinical benefit at first response assessment. Using this cutoff, patients with low sMICB (n = 23) had a CBR of 82.6% while patients who had high sMICB (n = 17) had a CBR of 29.4% (*p* = 0.001).

For survival analyses, patients were stratified by the median value of sMICB at PRE (611.08 pg/mL) and at FR (512.27 pg/mL). In univariate Cox regression analysis, high sMICB levels at FR were associated with worse PFS (Table 4). Patients with low sMICB levels (>median) at FR had a median PFS of 12.53 months (CI 95%: 1.55–23.50) versus a median PFS of 3.83 months (CI 95%: 2.22–4.11) for patients with high sMICB levels (*p* = 0.019) (Figure 6A). Furthermore, patients with low sMICB levels tended to have better OS (34.5 vs. 12.50 months) (Appendix A). Multivariate analysis on PFS including all clinicopathological features and biomarkers confirmed the independent role of sMICB, with a hazard ratio (HR) of 2.57 (CI 95%: 1.22–5.40; *p* = 0.012). No significant results were found at PRE with median value (Appendix A).

Patients were stratified considering the 75th percentile of sMICB at PRE (943.80 pg/mL) and at FR (825.87 pg/mL). In univariate Cox regression analysis, low sMICB levels at PRE and FR were also associated with improved PFS and OS (Table 4). At PRE, patients with sMICB levels below the 75th percentile had a median PFS of 10.53 (CI 95%: 3.21–17.84) and median OS of 28 months (CI 95%: 18.38–37.61), versus a median PFS of 2.57 months (CI 95%: 0.73–4.40) and median OS of 6.3 months (CI 95%: 0.1–12.8) for patients with sMICB levels above the 75th percentile (*p* = 0.013 and *p* = 0.023, respectively) (Figure 6B,C). At FR, patients with sMICB levels below the 75th percentile had a median PFS of 10.53 (CI 95%: 0.1–21.26) and median OS of 31.27 months (CI 95%: 20.98–41.55), versus median PFS of 2.53 (CI 95%: 1.8–3.66) and median OS of 9.7 months (CI 95%: 2.37–17.029) for patients with sMICB levels above the 75th percentile (*p* = 0.001 and *p* = 0.011, respectively) (Figure 6D,E).

#### 2.4.3. sGAL-3

In terms of response analysis, at FR, median sGAL-3 levels were significantly higher in patients with non-DCB with a median value of 10,297.75 pg/mL (IQR: 8105.84–14,298.70) compared to 9801.43 pg/mL (7941.03–11,820.17) in patients with DCB (*p* = 0.03) (Figure 7A). However, there was no statistical difference in sGAL3 levels measured at PRE in patients with DCB compared to patients with non-DCB, neither at PRE nor at FR in patients who were responders compared to non-responders (Appendix A).

Survival analysis showed that patients with high sGAL-3 levels (≥median) at PRE were associated in univariate Cox regression analysis with worse PFS and OS (Table 4). Kaplan–Meier analysis also showed a significant association of sGAL-3 at PRE with patient prognosis. Patients with high sGAL-3 levels (>median) had shorter PFS (4.47 vs. 12.53 months, *p* = 0.031) and OS (11.53 vs. 29.80 months, *p* = 0.049) (Figure 7B,C). The independent role of sGAL-3 at the FR of PFS and OS was confirmed in multivariate analysis including all clinicopathological features and biomarkers, with a hazard ratio (HR) of 2.45 (95% CI: 1.14–5.25; *p* = 0.021) and 4.915 (95% CI: 1.89–12.73; *p* = 0.001), respectively. No significant results were found for sGAL-3 levels at FR (Appendix A). In this case, as high levels of sGAL3 at FR led to non-DCB, we attempted to be more restrictive in survival analysis by considering the 75th percentile as cutoff (13,004.53 pg/mL). In univariate Cox regression analysis, low sGAL-3 levels at first response assessment were also associated with improved progression-free survival (Table 4). Patients with sGAL-3 levels below the 75th percentile had a median PFS of 8.87 (95% CI: 2.69–15.04) versus a median PFS of 2.60 months (95% CI: 0.1–6.87) for patients with sGAL-3 levels above the 75th percentile (*p* = 0.026) (Figure 7D).

Interestingly, patients with increased (>double) sGAL-3 levels at FR had shorter OS in comparison with patients harboring stable or decreased levels of sGAL-3 at FTE (3.9 vs. 22.6 months, *p* = 0.033) (Figure 8). Multivariate analysis including all clinicopathological variables (gender, age, tumor node metastasis (TNM) staging, smoking status, and the evolution of the rest of the biomarker levels between PRE and FR) confirmed that stable or decreased sGAL-3 levels at FR were independently associated with longer OS with an HR at 2.147 (95% CI: 1.046–4.409; *p* = 0.037). Changes in the rest of the biomarker levels between PRE and FR did not predict patients’ outcomes.

#### 2.4.4. sICOSL, sGAL1, sFGLF1, and sMICA

There was no impact of sICOSL, sGAL1, sFGFL1, and sMICA levels and variations on ORR, clinical benefit, PFS, or OS at any time of analysis (Appendix A).

## 3. Discussion

In our study, the main objective is to search for effective biomarkers from minimally invasive samples, such as soluble immune proteins. Despite the development of ICI-based therapies in advanced NSCLC, there is an urgent need for new biomarkers for patient selection and treatment optimization. In spite of the existence of three current biomarkers (PD-L1 expression, microsatellite instability (MSI), or tumor mutational burden (TMB)) in clinical practice approved by the FDA for selecting patients receiving immunotherapy, all of them currently have weaknesses [16,35,36]. Other approaches such as TCR repertoire were also proposed to predict the efficacy of anti-PD1 therapy in NSCLC. However, the results are still too preliminary to implement in routine clinical practice [37,38]. Therefore, there is a great need for more effective biomarkers, also outside the selection of patients who would benefit from treatment. Nowadays, liquid biopsy, compared with tissue biopsy, is a less invasive methodology for testing biomarkers, being specifically useful when there is insufficient tumor tissue or when the tumor is inaccessible. [39,40]. Plasmatic biomarkers have several advantages such as being easily accessible, allowing for sequential analysis during follow-up, and reflecting different tumor clones. Some studies on plasmatic proteins as biomarkers associated with immune checkpoint inhibitor (ICI) efficacy in NSCLC have been performed [28,41,42]. However, studies with plasmatic biomarkers for pembrolizumab optimization currently remain sparse.

This study is of great relevance, as we investigated the value of seven circulating immune proteins in plasma—sFGFL1, sCD276, sICOSL, sGAL-1, sGAL-3, sMICB, and sMICA—before pembrolizumab treatment (PRE) and at first response assessment (FR) in advanced NSCLC patients. As far as we know, this study reports, for the first time, novel potential diagnostic, predictive, and prognostic plasmatic immune-related biomarkers in advanced NSCLC patients treated with pembrolizumab as a first-line treatment. In this study, we comprehensively analyzed the circulating immune proteins at two different time points, baseline and first response assessment, which allowed for dynamic analyses. It is also important to focus on the study of plasmatic immune-related biomarkers at first response assessment because the immunotherapy used with these patients could lead to changes in the tumor microenvironment, including the secretion of the proteins studied in this work. Other studies have only focused on baseline analysis. For instance, levels of sPD-L1 and sGranB before treatment were associated with outcomes in NSCLC treated with nivolumab [28], high plasma sPD-L1 levels at pretreatment were associated with poor prognosis in patients with advanced lung cancer [41] and sPD-1 and sPD-L1 at baseline could predict nivolumab efficacy in NSCLC patients [42].

We observed that former or current smokers exhibit higher levels of sICOSL, sCD276, and sMICA than nonsmokers. These three molecules are immunosuppressive proteins involved in the tumor microenvironment. Tobacco has an immunosuppressive effect on the tumor microenvironment, as previously reported [43,44,45]. In agreement with our results, Inamura et al. revealed, in a cohort of 270 lung adenocarcinomas, that high CD276 expression is correlated with former or current smokers and associated with decreased survival [46]. No other correlations were previously described for ICOSL and MICA and tobacco history.

FGL1 is a novel ligand of LAG3 that results in T-cell depletion and subsequent dysfunction, as well as tumor cell escape from immune surveillance [47]. FGL1 is upregulated in tumor tissues (including lung, prostate, melanoma, colorectal, breast, and brain tumors) based on meta-analysis of the Oncomine databases [47]. Our results confirmed that sFGL1 was raised in patients with advanced NSCLC compared to controls with no cancer and has the potential to be a useful biomarker in plasma for the diagnosis of NSCLC. In line with our results, Wentao Li. et al. [48] revealed that fibrinogen-like protein 1 is normally expressed in the pneumonia group (n = 10) but was upregulated in the advanced lung adenocarcinoma group (n = 7). Compared to our study, the number of samples used in this study was too small, which implies low statistical power. Furthermore, this research group used isobaric tags for relative and absolute quantification (iTRAQ) labeling coupled with multidimensional liquid chromatography–tandem mass spectrometry iTRAQ-coupled 2D LC-MS/MS, a technology which is extremely laborious, time-consuming, and very expensive [49]. Contrary to these authors, we used Luminex^®^ MAP technology, a faster technology that allows for higher throughput, smaller sample volume, and higher sensitivity (≥) [50].

GAL-1 is a member of a family of β-galactoside-binding proteins with immunosuppressive molecules expressed by many types of cancer [51]. Our results revealed that sGAL-1 was elevated in patients with advanced NSCLC treated with pembrolizumab compared to controls with no cancer. This may provide information about the diagnosis of NSCLC. No studies on lung cancer have been reported before; however, other diagnostic analyses have been performed on other types of cancer. As supported by our results, galectin-1 expression in tissue specimens from thyroid cancer patients could be a reliable diagnostic marker for thyroid carcinomas [52]. Moreover, Navarro et al. reported that the detection of GAL-1 circulating levels shows strong potential for use as a novel biomarker for the diagnosis of pancreatic ductal adenocarcinoma (PDA) patients [53].

GAL-3 is another relevant member of a family of β-galactoside-binding proteins that has emerged as an important regulator of different functions important in cancer biology including apoptosis, metastasis, immune surveillance, molecular trafficking, mRNA splicing, gene expression, and inflammation [54]. In NSCLC, high GAL-3 expression in tumor cells is related to tumor progression and poor prognosis [55,56]. Our results revealed that sGAL-3 is useful not only to assess the immune response, but also to predict clinical outcomes for immunotherapy in NSCLC. Specifically, high levels of sGAL-3 at FR could predict nonclinical benefit and poor progression-free survival (PFS), while high levels of sGAL-3 at PRE are predictors of worse PFS and overall survival (OS). Additionally, variation in sGAL-3 between baseline and first response assessment impacted OS. We showed that a decrease in sGAL-3 after four cycles of pembrolizumab was associated with an improvement in OS. Assessing sGAL-3 kinetics between PRE and FR reflects the impact of pembrolizumab on the biomarker’s production or destruction and may be helpful to identify non-responders before the first radiological evaluation, which usually occurs after four to six cycles. Recently, Jun Sum Kim et al. studied the role of sGAL3 in NSCLC receiving ICIs, with similar results showing that patients with higher sGAL3 levels at baseline in serum or plasma had worse OS (n = 56) [57]. However, patients did not have the first response assessment measurement and no dynamic analysis was performed. Furthermore, the type of sampling (plasma or serum) was heterogeneous, which may imply nonhomogeneous results between samples.

We also reported that sGAL-3 could be a good diagnostic biomarker for NSCLC alone or in combination with sFGL1. High levels of sGAL-3 and sFGL1 were related with patients with NSCLC. Conversely, controls exhibited low levels of these analytes. No studies were found about the value of GAL-3 as a diagnostic biomarker in lung cancer. However, some groups have reported that serum galectin-3 is higher in cases with pancreatic carcinoma than in benign pancreatic diseases and healthy subjects [58], which has also been observed in heart and kidney diseases [59,60]. Furthermore, patients with metastatic prostate cancer had higher levels of serum galectin-3 compared with control subjects without cancer [61]. Serum GAL-3 levels were significantly increased in breast cancer patients compared with healthy control subjects [62]. Moreover, we showed that the combination of sFGL1 and sGAL-3 resulted in improved diagnostic efficacy for cancer patients. Both analytes are ligands of lymphocyte activation gene 3 (LAG-3), one of the most promising immune checkpoints next to PD-1 and CTLA-4. We hypothesize that tumors secrete high levels of sFGLF1 and sGAL-3 as a mechanism of immunoevasion through the activation of LAG-3, an immune checkpoint inhibitor that exerts inhibition on T-cell activation.

CD276, also known as B7-H3, is a novel immune checkpoint from the B7 family that is highly expressed in NSCLC [63]. Initially, B7-H3 was thought to co-stimulate immune response; however, recent studies have shown that it has a coinhibitory effect on T-cells, contributing to tumor cell immune evasion [64]. Soluble CD276 (sCD276) is produced by alternative splicing from the intron of B7-H3 [65] or matrix metallopeptidase (MMPS) [66]. We found that patients who presented with an objective response to pembrolizumab had significantly lower sCD276 levels at pembrolizumab initiation than patients who were non-responders. This could reflect the activation of the T-cell immune response, which is known to be associated with ICIs’ better response. When analyzing the prognosis, we found a tendency favoring patients with low levels of sCD276 for OS and PFS. The prognostic value of sCD276 has already been explored in a few cancer studies. It has been reported that high levels of sCD276 are associated with unfavorable prognosis in ovarian cancer patients and gastric adenocarcinoma patients, which conforms with our results [11,67]. To date, only one recent study reported its involvement in the response to ICI in NSCLC. In contrast to the results seen in our study, these authors noticed that high levels of sCD276 were associated with better outcomes [68]. Despite the bigger cohort of patients used, we employed Luminex technology, which is more sensitive than ELISA methodology. Moreover, our results are concordant with studies, presented in a review, performed on tissue specimens in which CD276 expression was correlated with poor prognosis, which supports the validity of our results [69].

MICB, known as part of the stimulatory natural killer group 2-member D (NKG2D) ligand, is a polymorphic protein that is induced upon stress, damage, or transformation of cells that acts as a ‘kill me’ signal through the natural killer group 2-member D receptor expressed in cytotoxic lymphocytes. The soluble isoform of MICB in the bloodstream is derived from alternative splicing, PI-PLC-mediated cleavage, proteolytic shedding, or via exosome secretion [70]. It has been reported, contrary to membrane-bound form, that the soluble form of MICB induces downmodulation of NKG2D expression on systemic and tumor-infiltrated NK and T-cells and thus results in its functional impairment [71,72,73,74]. Our results suggest that sMICB (measured both at PRE and FR) may play an important role in evaluating diagnosis, survival, and clinical benefit under pembrolizumab, with this study being of great clinical relevance in lung cancer. In our study, patients with NSCLC had high levels of sMICB compared to controls. Moreover, high levels of sMICB at PRE and at FR were associated with worse prognosis. In accordance with our results, Tamaki et al. [75] revealed that sMICB serum levels are significantly increased in 60 stage IV oral squamous cell carcinoma (OSCC) patients compared to healthy controls and are significantly associated with decreased survival rates (OS and PFS). Bao-Jin Wu et al. [76] also reported that elevated sMICB serum levels were associated with poor overall and progression-free survival in 125 malignant melanoma patients at different stages of the disease. More recently, the absence of sMICB in baseline serum of stage III and IV melanoma patients treated with immune checkpoint blockades correlated with improved survival [77]. To the authors’ knowledge, this is the first study to determine the significance of sMICB in advanced NSCLC patients both at PRE and FR. Focusing on the difference between these two moments, our results revealed more significance in the results of sMICB measured at first response assessment than baseline. This led us to think that when patients were treated with pembrolizumab, cells try to activate other inhibitory pathways in the tumor microenvironment such as the release of sMICB.

The relation of high levels of sMICB with poor prognosis could be explained because the shedding of sMICB in the bloodstream causes the ineffectiveness of NKG2D-mediated immunity in epithelial cancer including lung, ovarian, colon, breast, neuroblastoma, melanoma, and prostate cancer [74]. Our findings suggest the utility of sMICB levels as a marker for predicting patients’ immune response and clinical outcomes in advanced NSCLC patients treated with pembrolizumab. Due to tumor cells’ cleavage of the MICB from the cell membrane for immune escape leading to progression and poor prognosis, we hypothesize that targeting sMICB could be a basis for new therapeutic approaches for these patients. A humanized IgG1 monoclonal antibody that binds to MICB, CLN-619, is currently being evaluated alone and in combination with pembrolizumab for the treatment of patients with advanced solid tumors (NTC05117476) [78].

When we analyzed potential plasmatic immune-related biomarkers to predict durable clinical benefit, sMICB followed by sGAL3 seem to be the best in our cohort of 52 advanced NSCLC patients. Furthermore, sCD276 is also proposed as a good biomarker for predicting patients who respond to pembrolizumab. All of these molecules, as previously explained, in their soluble form, act by inducing immunosuppression in the tumor immune microenvironment: MICB prevents NK cells from performing their functions, GAL3 induces T-cell apoptosis and TCR downregulation, and CD276 inhibits the function and survival of T-cells. Consequently, a microenvironment characterized by increased immunosuppression is present, leading to a lack of response to immunotherapy in patients with elevated levels of these analytes. Regarding prognostic value analysis, high levels of sMICB and sGAL-3 were associated with poor prognosis, with these analytes being the best predictors of survival in these patients. These two plasmatic immune-related biomarkers are well-defined immunosuppressive circulating proteins that favor a poor prognosis in advanced NSCLC patients. Most other studies performed in NSCLC about circulating immune proteins have focused on sPD-L1, with contradictory results [28,79]. Other soluble analytes such as sGranB have also been associated with prognosis; however, no correlations with response to immunotherapy were found [28]. Tiako Meyo et al. also studied the predictive value of PD-1, PD-L1, VEGFA, CD40 ligand, and CD44 for nivolumab therapy in advanced non-small cell lung cancer, which revealed a potential predictive role of baseline sCombo (sPD-1 and/or sPD-L1 expression) for nivolumab efficacy [42]. No other studies have been performed on lung cancer with the circulating immune proteins included in this study.

Despite all these exciting results, our study has some limitations. Although only a small cohort of patients was analyzed, we observed significant results regarding objective response, clinical benefit, and survival. Finally, to confirm the predictive and prognostic value of sMICB, sGAL-3, and sCD276, it would be necessary to use a validation cohort. This study also has several strengths. First, we used a prospective cohort of patients with plasma samples previously well characterized that underwent rigorous preanalytical conditioning. Second, our study presents two different samples at baseline and first response assessment for each patient, which allowed for dynamic analyses. Third, we employed an ultrasensitive multiplex methodology that increased sensitivity and reduced costs, time, and samples used. Moreover, we used a robust immunoassay, so the possibility of assay-dependent variability was not addressed. Studies using liquid biopsy, as demonstrated in the present study, indicate that we should continue the search for new immune biomarkers that can better assist us in selecting patients who will benefit from immunotherapy.

## 4. Materials and Methods

### 4.1. Study Population

Between 2018 and 2021, a total of 52 advanced NSCLC patients were treated with first-line pembrolizumab in monotherapy (200 mg every 21 days) and a total of 34 controls with no cancer (16 healthy subjects, 9 with chronic obstructive pulmonary disease, and 9 with cystic fibrosis) were enrolled in this study. In brief, all cases were individuals with histologically confirmed NSCLC and those with autoimmune disease or acquired immunodeficiency syndrome (AIDs) were excluded from the present study. Patients’ clinical data (demographic and clinicopathological) were collected from medical records. Follow-up (until May 2023) was carried out according to the institutional standard for advanced NSCLC patients treated with ICIs.

This study was conducted in accordance with the ethical code of the World Medical Association (Declaration of Helsinki), and the protocol was approved by the ethical review board of the General University Hospital of Valencia. All subjects provided their informed consent for inclusion before participating in this study.

### 4.2. Sample Collection

Peripheral blood samples were collected in EDTA-containing tubes at baseline (PRE) before the first administration of pembrolizumab for all NSCLC patients (N = 52) and at the first response assessment (FR) (N = 42). Plasma was separated by centrifugation (4000 rpm. 10 min, 4 °C) within 2 h after blood collection and immediately stored at −80 °C until further processing. For this study, plasma samples were defrosted and centrifuged (1000 rpm, 10 min) before the immunoassay experiment.

### 4.3. Multiparametric Immunoassay Based on Luminex xMAP

Plasma levels of inducible T-cell costimulatory ligand (sICOSL), cluster of differentiation 276 (sCD276), fibrinogen-like protein 1 (sFGL1), galectin-1 (sGAL-1), human galectin-3 (sGAL-3), human MHC class I polypeptide-related sequence A (sMICA), and human MHC class I polypeptide-related sequence B (sMICB) were assayed using a multiparametric immunoassay commercial kit, MILLIPLEX^®^ Human Immuno-Oncology Checkpoint Protein Panel 2—Immuno-Oncology Multiplex Assay, Cat #HCKP2-11K, according to manufacturer’s instructions based on Luminex xMAP technology (Luminex Corp, Austin, TX, USA). Quality controls (QC1 and QC2) and a calibration curve based on 1:4 dilutions of the highest standard were used for quantification and as internal controls for intra- and interassay reproducibility. In brief, 25 µL of plasma (diluted 1:2) was used for each sample and mixed with proper regents and the mixture of monoclonal antibodies (sICOSL, sCD276, sFGL1, sGAL-1, sGAL-3, sMICA, and sMICB). These monoclonal antibodies are covalently bound to the surface of magnetic microspheres dyed with accurate amounts of red and infrared fluorophores in order to produce a single spectral signature of each one, which can be detected using the Luminex platform (Luminex Corp, Austin, TX, USA). Protein quantifications were determined by the fluorescently labelled secondary antibody whose signal intensity is proportional to the detected analyte concentration. The fluorescent signal of all samples was read on a Luminex 100/200™ instrument (Luminex Corp). Based on the measurements of 7 diluted standard concentrations provided by the manufacturer, a five-parameter standard curve was used to convert optical density values into concentrations (pg/mL). Data for a minimum of 50 beads per cytokine were collected for each standard and sample. The final concentrations (expressed in pg/mL) were calculated using BelysaTM software version 1.2 (Merck Millipore, Billerica, MA, USA). All interassay and intra-assay coefficients of variation (CV) were below 15%. The lower limit of quantification (LLOQ) of sICOSL, sCD276, sFGL1, sGAL-1, sGAL-3, sMICA, and sMICB was 12.2 pg/mL, 195 pg/mL, 48.8 pg/mL, 61 pg/mL, 48.8 pg/mL, 12.2 pg/mL, and 104 pg/mL, respectively.

### 4.4. Exploratory Endpoints Patients Evaluation

The relationship between plasma levels of biomarkers and tumor response and survival was to be explored. To this end, tumor response was evaluated every 21 days using the Response Evaluation Criteria in Solid Tumors version 1.1 (RECIST 1.1). The objective response rate (ORR) was evaluated and defined as the proportion of patients achieving complete (CR) or partial response (PR), stable disease (SD), or progressive disease (PD). Durable clinical benefit (DCB) (i.e., complete response, partial response, or stability, lasting 6 months or more after initiation of pembrolizumab treatment) and non-DCB (PD within 6 months after treatment start) were also analyzed. Progression-free survival (PFS) was described as the interval from the beginning of pembrolizumab treatment to the objective disease progression or last follow-up. Overall survival (OS) was defined as the interval from the beginning of pembrolizumab treatment to death or last follow-up.

### 4.5. Statistical Analysis

PRE and FR samples were compared using Wilcoxon signed-rank test. The comparison of median soluble levels of all tested biomarkers between any groups was performed using a nonparametric Mann–Whitney U test (two groups) and Kruskal–Wallis (more than two groups) to compare continuous variables. The association between discrete variables was evaluated using X2 tests. Graphs comparing metrics across groups show the median and the interquartile range (IQC) because data are not normally distributed with the Kolmogorov–Smirnov test.

Receiving operating curve (ROC) method was used to determine a cutoff level for each biomarker with a significant difference for DCB and ORR. ROCs were also used for evaluating the diagnostic power of biomarkers. Other predictive parameters were also evaluated, including sensitivity, specificity, cutoff value, positive predictive value, negative predictive value, and area under the ROC (AUC) with a 95% confidence interval (CI), to assess the discrimination power of individual biomarkers. The identification of the cut-point value requires a concurrent evaluation of sensitivity and specificity. One of the commonly employed methods is the Youden index method [80]. This approach defines the optimal cutoff point as the point that maximizes the Youden function, which represents the difference between the true positive rate and the false positive rate across all potential cutoff values. In general, an AUC of 0.5 suggests no discrimination, 0.7 to 0.8 is considered acceptable, 0.8 to 0.9 is considered excellent, and more than 0.9 is considered outstanding.

Kaplan–Meier method was used to determine PFS and OS. Patients were stratified by the median value and the 75th percentile. Comparison between survival curves was performed using log-rank method. Hazard ratio (HR) and 95% confidence interval (95% CI) were calculated using the univariate Cox proportional hazards regression model. Multivariate analysis was performed using Cox proportional hazards model (PFS and OS). All significant variables from the univariate analyses were entered into the multivariate analyses in a forward stepwise Cox regression analysis. Statistical analyses were performed using the Statistical Package for the Social Sciences (SPSS, Chicago, IL, USA) version 23.0. *p*-values were considered significant if <0.05.

## 5. Conclusions

In advanced stages, acquiring tissue samples can be challenging in many cases. We must find other alternatives for the study of biomarkers; therefore, our study focused on identifying minimally invasive biomarkers. We aimed not only to improve diagnosis but also to forecast clinical outcomes in advanced non-small cell lung cancer (NSCLC). Our groundbreaking study marks the first comprehensive exploration of the diagnostic, prognostic, and predictive significance of seven different soluble mediators involved in immunoregulation in plasma samples from advanced NSCLC patients with first-line pembrolizumab treatment, both at baseline (PRE) and first response assessment (FR). Firstly, we observed significant associations between sICOSL, sMICA, and sCD276 levels and smoking status. Secondly, our findings highlight the potential diagnostic utility of plasma levels of sFGL1, sGAL-3, and sGAL-1 in discerning advanced NSCLC. Finally, our study unveiled the association of sMICB, sCD276, and sGAL3 with clinical outcomes, suggesting a pivotal role in evaluating response, survival, and clinical benefit under pembrolizumab therapy. The simultaneous analysis of these markers using this multiparametric approach provides valuable clinical insights and presents an opportunity to streamline cost and time considerations in prognostic evaluations. Novel results that have not previously been described in advanced NSCLC were obtained using liquid biopsy. However, validating these results with an external cohort and even analyzing more soluble proteins would be necessary to strengthen these findings. Altogether, these novel findings position our research as significant progress in the clinical management of lung cancer patients through liquid biopsy and provide promising possibilities for enhanced individualized care and treatment strategies.

## Figures and Tables

**Figure 1 ijms-24-17587-f001:**
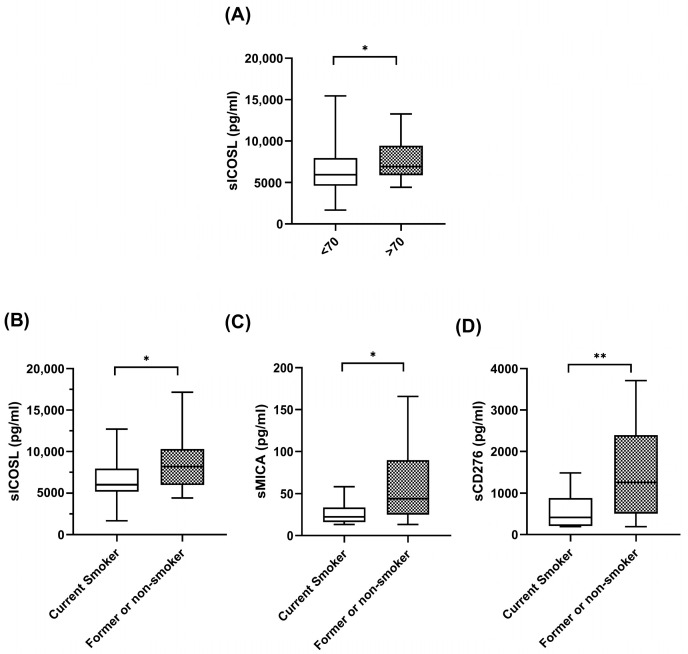
Correlations between soluble biomarkers and clinicopathological variables. (**A**) sICOSL levels at baseline in patients < 70 years (n = 33) and patients > 70 years (n = 19). (**B**–**D**) sICOSL, sCD276, sMICA at baseline in patient current smokers (n = 37) and patients former or nonsmokers (n = 15). The bold horizontal lines in the boxplots are medians and bars represent minimum and maximum values. n: sample size. * *p* < 0.05; ** *p* < 0.01.

**Figure 2 ijms-24-17587-f002:**
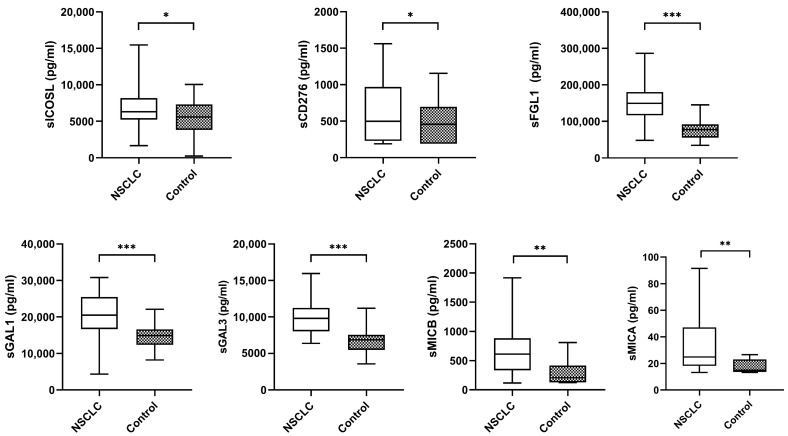
Levels of plasma biomarkers between NSCLC samples and controls. The bold horizontal lines in the boxplots are medians and bars representing minimum and maximum values. *p*-values were calculated using the Mann–Whitney test. Asterisks indications: * *p* < 0.05; ** *p* < 0.01; *** *p* < 0.001. NSCLC: non-small cell lung cancer.

**Figure 3 ijms-24-17587-f003:**
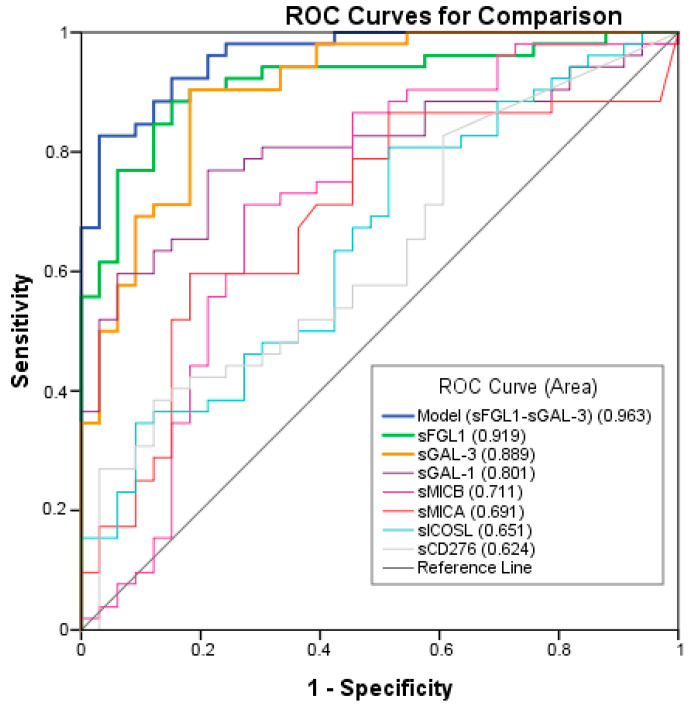
Receiver operating characteristic (ROC) curves of individual or combination of Model, sFGL1, sGAL-3, sGAL-1, sMICB, sMICA, sICOSL, sCD276 plasma tumor biomarkers in advanced NSCLC compared to the controls.

**Figure 4 ijms-24-17587-f004:**
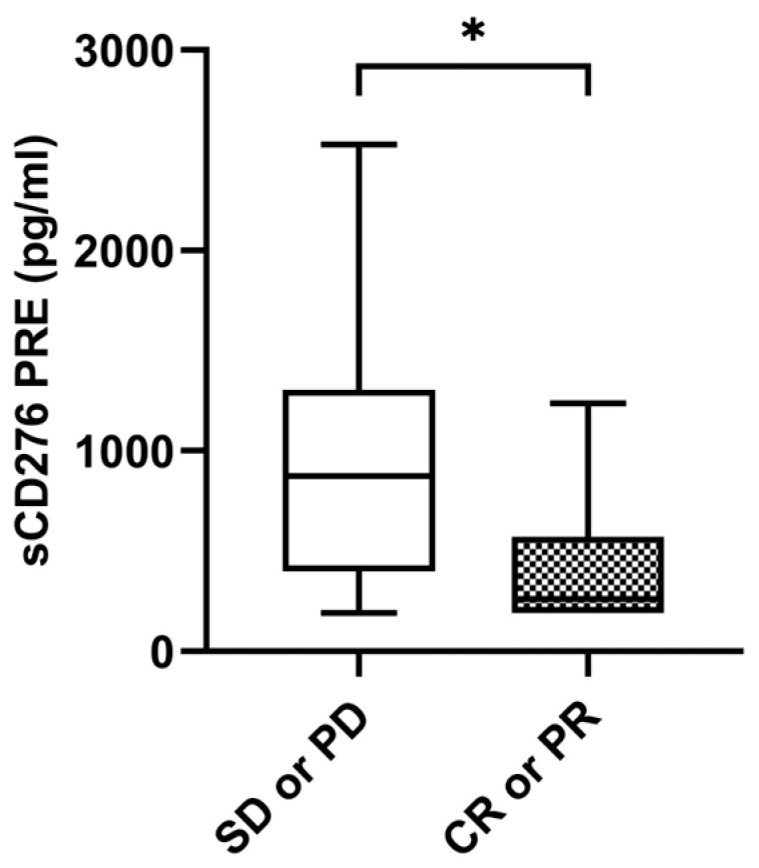
sCD276 and tumor response. sCD276 levels at PRE in patients with tumor response (n = 21) and patients without tumor response (n = 31). The bold horizontal lines in the boxplots are medians and bars represent minimum and maximum values. *p*-values were calculated using the Mann–Whitney test. Asterisks indication: * *p* < 0.05. SD: stable disease; PD: progression; PR: partial response; CR: complete response; PRE: baseline.

**Figure 5 ijms-24-17587-f005:**
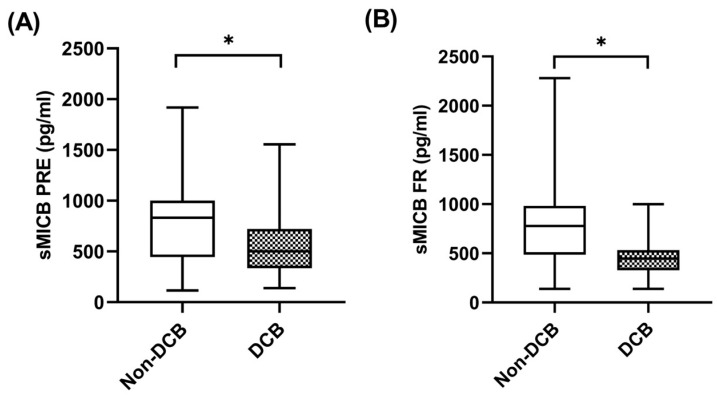
sMICB and clinical benefit. (**A**) sMICB levels at PRE in patients with non-DCB (n = 23) and patients with DCB (n = 29). (**B**) sMICB levels at FR in patients with non-DCB (n = 16) and patients with DCB (n = 24). The bold horizontal lines in the boxplots are medians and bars represent minimum and maximum values. Horizontal lines in the boxplots are medians and bars represent minimum and maximum values. *p*-values were calculated using the Mann–Whitney test. Asterisks indication: * *p* < 0.05. DCB: durable clinical benefit; PRE: baseline; FR: first response assessment.

**Figure 6 ijms-24-17587-f006:**
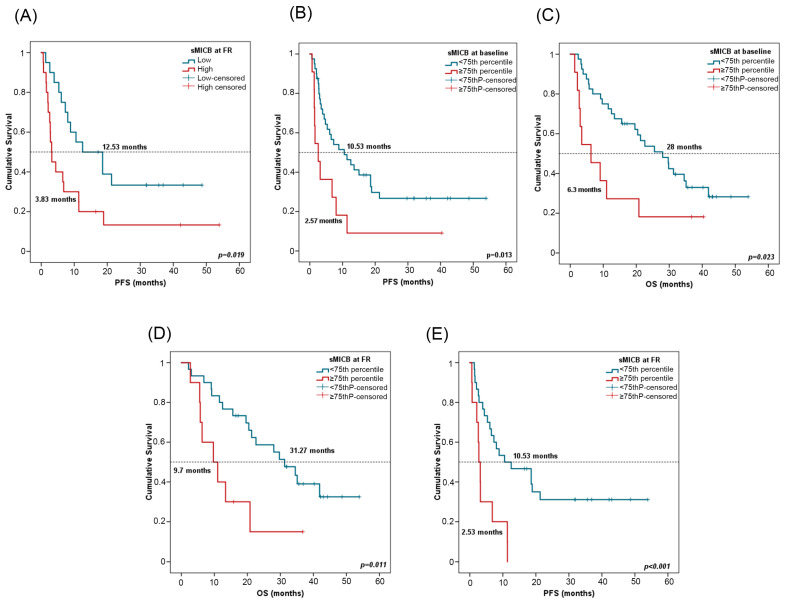
sMICB and progression-free survival (PFS) and overall survival (OS). (**A**) PFS in line with sMICB levels at FR. Cutoff values correspond to the median soluble levels. Red lines represent patients with high levels of sMICB (n = 20), whereas blue lines represent patients with low levels of sMICB (n = 20). (**B**,**C**) PFS and OS in compliance with sMICB levels (above (n = 11) or below (n = 40) the 75th percentile) at PRE. (**D**,**E**) PFS and OS according to sMICB levels (above (n = 10) or below (n = 30) the 75th percentile) at FR. *p*-values were calculated using the log-rank test. FR: first response assessment; PRE: baseline; DCB: durable clinical benefit; non-DCB: nondurable clinical benefit.

**Figure 7 ijms-24-17587-f007:**
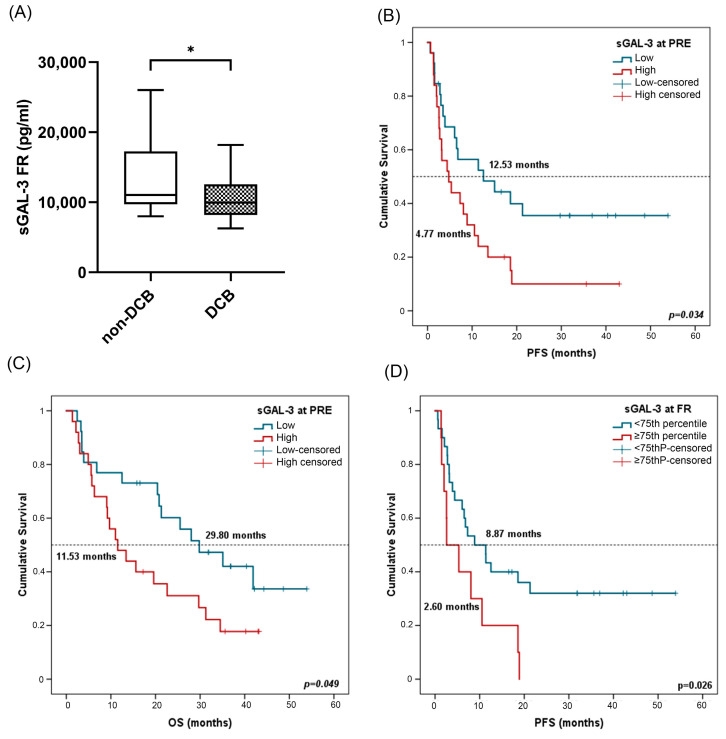
sGAL-3 and clinical benefit, progression-free survival (PFS), and overall survival (OS). (**A**) sGAL-3 levels at FR in patients with non-DCB (n = 16) and patients with DCB (n = 25). The bold horizontal lines in the boxplots are medians and bars represent minimum and maximum values. (**B**,**C**) PFS and OS based on sGAL-3 levels at PRE. Cutoff values correspond to the median soluble levels. Red lines represent patients with high levels of sGAL-3 (n = 25), whereas blue lines represent patients with low levels of sGAL-3 (n = 26). (**D**) PFS according to sGAL-3 levels (above or below the 75th percentile) at FR. Red lines represent patients with high levels of sGAL-3 (above the 75th percentile) (n = 10), whereas blue lines represent patients with low levels of sGAL-3 (below the 75th percentile) (n = 30). P-values were calculated using the Mann–Whitney test (**A**) or log-rank test (**B**–**D**). Asterisks indication: * *p* < 0.05. FR: first response assessment; PRE: baseline; DCB: durable clinical benefit; non-DCB: nondurable clinical benefit.

**Figure 8 ijms-24-17587-f008:**
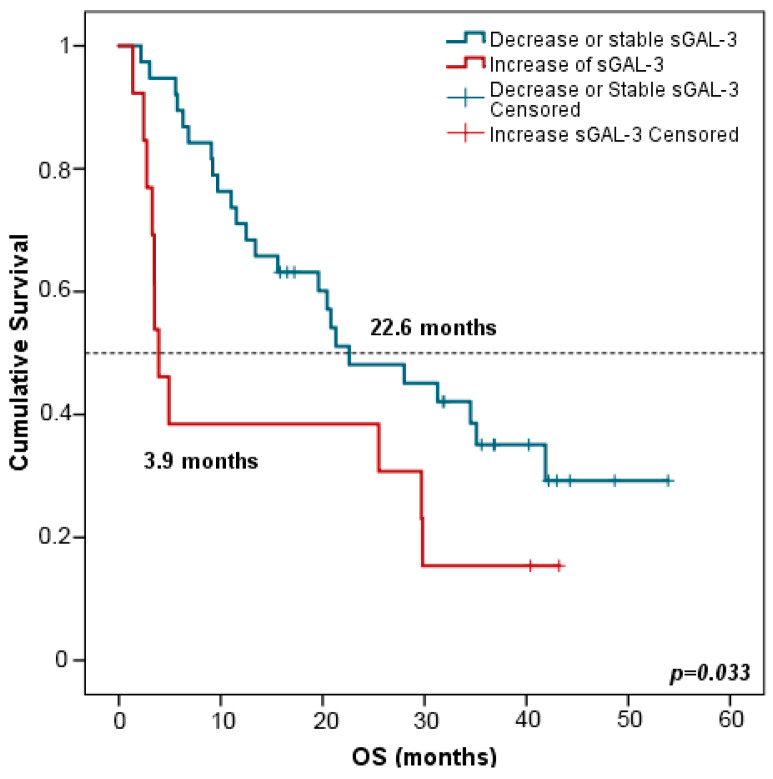
Kaplan–Meier survival curves based on the evolution of sGAL-3 concentrations between PRE and FR. OS stratified in decreased or stable sGAL3 levels (n = 38) vs. increased sGAL-3 levels (n = 13). Blue lines represent patients with decreased or stable levels of sGAL3 whereas red lines represent patients with increased levels of sGAL3. *p*-values were obtained using the log-rank test. OS: overall survival; FR: first response assessment; PRE: baseline.

**Table 1 ijms-24-17587-t001:** Clinicopathological characteristics of NSCLC cohort.

Patient Characteristics	Patients	
	*n* = 52	%
Age at diagnosis (median, range)	67.5 [IQR 51–89]	
Gender		
Male	39	75
Female	13	25
Histology		
ADC	34	65.4
SCC	13	25
Others	5	9.6
Stage		
III	12	23.1
IVA	13	25
IVB	27	51.9
Performance status		
0–1	44	84.6
2	7	13.5
Smoking status		
Current	37	71.2
Former	11	21.2
Never	4	7.7
Progression		
Yes	38	73.1
No	14	26.9
Exitus		
Yes	36	69.2
No	16	30.8

*n*: number; ADC: adenocarcinoma; SCC: squamous cell carcinoma; IQR: interquartile range.

**Table 2 ijms-24-17587-t002:** Median levels of soluble analytes measured using Luminex technology.

Analyte	Median at PRE (pg/mL)	IQR at PRE (pg/mL)	Median at FR (pg/mL)	IQR at FR (pg/mL)
sICOSL	6413.32	259.88–8220.46	6411.42	5336.64–9826.69
sCD276	611.78	238.45–1247.62	885.78	231.51–1749.88
sFGL1	151,042.76	121,193.97–189,434.59	156,594.82	125,794.24–211,091.35
sGAL-1	20,506.60	16,678.27–25,437.53	21,408.75	18,321.44–26,125.93
sGAL-3	9991.40	8066.19–12,776.39	10,252.91	8728.84–13,004.53
sMICA	28.22	19.00–54.07	13.33	16.53–41.62
sMICB	611.08	343.44–929.92	512.27	393.76–825.87

IQR: interquartile range; PRE: baseline; FR: first response assessment.

**Table 3 ijms-24-17587-t003:** Diagnostic accuracies of plasma biomarkers of advanced NSCLC.

	AUC (95% CI)	Sensitivity	Specificity	PPV	NPV
NSCLC vs. Healthy					
sFGL1	0.919 (0.860–0.978)	0.885	0.853	0.902	0.828
sGAL-3	0.889 (0.817–0.960)	0.904	0.794	0.904	0.794
sGAL-1	0.801 (0.709–0.894)	0.596	0.941	0.939	0.604
sMICB	0.711 (0.592–0.830)	0.712	0.706	0.712	0.706
sMICA	0.691 (0.575–0.807)	0.538	0.818	0.824	0.529
sICOSL	0.651 (0.535–0.769)	0.808	0.471	0.700	0.615
sCD276	0.624 (0.504–0.743)	0.269	0.971	0.933	0.465
Model (sFGL1-sGAL3)	0.963 (0.929–0.996)	0.827	0.971	0.997	0.785

NSCLC: non-small cell lung cancer; AUC: area under the curve; CI: confidence interval; PPV: positive predictive value; NPV: negative predictive value.

**Table 4 ijms-24-17587-t004:** Results from the univariate Cox regression model for OS and PFS.

	PFS			OS		
	HR	95% CI	*p*-Value	HR	95% CI	*p*-Value
sMICB at FR (high vs. low)	2.348	1.129–4.883	0.022 *	1.827	0.838–3.983	0.129
sMICB at PRE 75th percentile	2.454	1.180–5.102	0.016 *	2.378	1.104–5.125	0.027 *
sMICB at FR 75th percentile	3.643	1.611–8.241	0.002 **	2.938	1.232–7.005	0.015 *
sGAL3 at PRE (high vs. low)	2.024	1.053–3.890	0.034 *	1.949	0.992–3.830	0.053
sGAL3 at FR 75th percentile	2.341	1.082–5.064	0.031 *	2181	0.955–4.980	0.064

PFS: progression-free survival; OS: overall survival; HR: hazard ratio; CI: confidence interval. FR: first response assessment; PRE: baseline * *p* < 0.05, ** *p* < 0.01.

## Data Availability

All data generated or analyzed during this study are included in this published article and its Appendix A files. Data that supports these findings are available from the corresponding authors upon reasonable request.

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
