# Peer review of "Circulating Immune Proteins: Improving the Diagnosis and Clinical Outcome in Advanced Non-Small Cell Lung Cancer"

_ijms, 2023, doi:10.3390/ijms242417587_

Round 1

Reviewer 1 Report

Comments and Suggestions for Authors

The authors that examined the new predictive biomarkers for immuno-checkpoint inhibitors (ICIs) response for Non-Small Cell Lung Cancer (NSCLC) cancer treatment in patients, which more benefit the management treatment in the clinical routine. With pembrolizumab treatment, they found that sFGL1, sGAL-3 and sGAL-1 allowed an optimal diagnostic efficacy for cancer patients. And if added the combination of sFGL1 and sGAL-3 significantly can improve diagnostic accuracy. It is seem important and interesting, some comment as following:

1.     The authors that provided a summarized cartoon in the whole concept.

2.     What soluble immune factors is how to define should be mention.

3.     In the patients with pembrolizumab treatment, PD-L1 level should be mention and it is how possible correlated to sFGL1, sGAL-3 and sGAL-1 will be discussed.

Reviewer 2 Report

Comments and Suggestions for Authors

The Manuscript ID: ijms-2656693 submitted to the “IJMS” entitled, “Soluble Immune Factors: improving the Diagnosis and Clinical Outcome in Advanced Non-Small Cell Lung Cancer “ was reviewed.

The authors studied the potential diagnosis, predictive, prognostic roles of seven soluble immune-related biomarkers, including sFGL1, sICOSL, sCD276, sGAL-1, SGAL-3, sMICA, and SMICB in patients with stage III and IV lung adenocarcinoma treated with pembrolizumab, and concluded that some of them have benefits in addition to currently approved tumor PD-LI expression, microsatellite instability, and total tumor mutation burden.

The following comments were provided. 

  1. The authors well described the current status of immune checkpoint inhibitors in lung adenocarcinoma, in the “Introduction” and “Discussion” sections. They discussed the results of each immune-related biomarker, mainly sFGL1, sGAL-1 and sCD276, and compared with the literatures in spite of different caners. But, the comparison among them and other less significant biomarkers was not discussed.
  2. The higher levels of immune biomarkers in NSCLC patients than the controls could not translate to the diagnostic value of biomarker clinically. It also appeared in other non-malignant diseases.
  3. Too many figures and tables made the manuscript not easily readable. Some tables or figures can be fused such as Table 3 and Figure 1, even replaced with data description only.
  4. I would like to suggest the authors to describe the results of predictive and prognostic values of the seven biomarkers separately. (line 175 to 328)
  5. The authors should reduce the use of abbreviations. Besides, the abbreviations, such as DCB (line 112) should be described in the first appearance.
  6. The legend of Figure 6 was not correct and overlapped with Figure 5.
  7. There were many grammar problems, unclear sentences and incorrect styles in the manuscript, such as line 82-84, line 115-116, line 182-185, line 191-192, line 275-278, etc.
  8. IC or CI (confidence interval) 95% ?
  9. Age at surgery (or diagnosis?) in Table 1
  10. Did the bar in the Figures 1, 2, 4, 5, 7 represent minimum and maximum values?
  11. RFS, relapse-free survival or PFS, progression-free survival in Table 5.
  12. The legend of Figure 7 is incorrect in line 312. The legend of Figure 8 is incorrect in line 322-324.
  13. “Table 4” in line 291 is incorrect.
  14. The authors did not discuss the differences between those measured at the baseline and at first response assessment and their meanings.
  15. The form of the journal should be followed. 

Comments on the Quality of English Language

The Manuscript ID: ijms-2656693 submitted to the “IJMS” entitled, “Soluble Immune Factors: improving the Diagnosis and Clinical Outcome in Advanced Non-Small Cell Lung Cancer “ was reviewed.

The authors studied the potential diagnosis, predictive, prognostic roles of seven soluble immune-related biomarkers, including sFGL1, sICOSL, sCD276, sGAL-1, SGAL-3, sMICA, and SMICB in patients with stage III and IV lung adenocarcinoma treated with pembrolizumab, and concluded that some of them have benefits in addition to currently approved tumor PD-LI expression, microsatellite instability, and total tumor mutation burden.

The following comments were provided. 

  1. The authors well described the current status of immune checkpoint inhibitors in lung adenocarcinoma, in the “Introduction” and “Discussion” sections. They discussed the results of each immune-related biomarker, mainly sFGL1, sGAL-1 and sCD276, and compared with the literatures in spite of different caners. But, the comparison among them and other less significant biomarkers was not discussed.
  2. The higher levels of immune biomarkers in NSCLC patients than the controls could not translate to the diagnostic value of biomarker clinically. It also appeared in other non-malignant diseases.
  3. Too many figures and tables made the manuscript not easily readable. Some tables or figures can be fused such as Table 3 and Figure 1, even replaced with data description only.
  4. I would like to suggest the authors to describe the results of predictive and prognostic values of the seven biomarkers separately. (line 175 to 328)
  5. The authors should reduce the use of abbreviations. Besides, the abbreviations, such as DCB (line 112) should be described in the first appearance.
  6. The legend of Figure 6 was not correct and overlapped with Figure 5.
  7. There were many grammar problems, unclear sentences and incorrect styles in the manuscript, such as line 82-84, line 115-116, line 182-185, line 191-192, line 275-278, etc.
  8. IC or CI (confidence interval) 95% ?
  9. Age at surgery (or diagnosis?) in Table 1
  10. Did the bar in the Figures 1, 2, 4, 5, 7 represent minimum and maximum values?
  11. RFS, relapse-free survival or PFS, progression-free survival in Table 5.
  12. The legend of Figure 7 is incorrect in line 312. The legend of Figure 8 is incorrect in line 322-324.
  13. “Table 4” in line 291 is incorrect.
  14. The authors did not discuss the differences between those measured at the baseline and at first response assessment and their meanings.
  15. The form of the journal should be followed. 

Reviewer 3 Report

Comments and Suggestions for Authors

The authors have analyzed the diagnostic, prognostic and predictive significance of 7 different soluble mediators involved in immunoregulation in plasma samples from 52 advanced NSCLC patients treated in first-line with pembrolizumab. They found sMICB, sCD276 and sGAL3 were associated with outcomes and may play an important part in evaluating the response, survival, and clinical benefit under pembrolizumab.

My comments:

1.     The author tried to present 7 plasma markers in one study report, but did not demonstrate any advantage for testing 7 markers at one time. They analyzed the data of each marker one by one, which made the paper too long and too complicated. It is difficult to read. Their conclusion also just pointed out the benefit of each marker, and not the strength for testing several markers at one time.

2.      The author should give the reasons why they chose the 7 markers in the introduction section first, to let the readers know what are these markers. It should not be described just in the discussion section. In the discussion, the author repeated a long description on the limitation of current immunotherapy as in the introduction section. It should be deleted.

3.     Although the authors found some interesting markers in this study, the case cohort was quite small. The criteria for defining the cut off for each marker were all different. Thus, it is quite difficult to remember.   

Comments on the Quality of English Language

The English should be reviewed by an English expert.
